# Enhancing Images with Coupled Low-Resolution and Ultra-Dark Degradations: A Tri-level Learning Framework

## ABSTRACT

Due to device constraints and lighting conditions, captured images frequently exhibit coupled low-resolution and ultra-dark degradations. Enhancing the visibility and resolution of ultra-dark images simultaneously is crucial for practical applications. Current approaches often address both tasks in isolation or through simplistic cascading strategies, while also relying heavily on empirical and manually designed composite loss constraints, which inevitably results in compromised training efficacy, increased artifacts, and diminished detail fidelity. To address these issues, we propose **TriCo**, the first to adopt a **Tri**-level learning framework that explicitly formulates the bidirectional **Co**operative relationship and devises algorithms to tackle coupled degradation factors. In the optimization across Upper (U)-Middle (M)-Lower (L) levels, we model the synergistic dependencies between illumination learning and super-resolution tasks within the M-L levels. Moving to the U-M levels, we introduce hyper-variables to automate the learning of beneficial constraints for both learning tasks, moving beyond the traditional trial-and-error pitfalls of the learning process. Algorithmically, we establish a Phased Gradient-Response (PGR) algorithm as our training mechanism, which facilitates a dynamic, inter-variable gradient feedback and ensures efficient and rapid convergence. Moreover, we present the Integrated Hybrid Expert Modulator (IHEM), which merges inherent illumination priors with universal semantic model features to adaptively guide pixel-level high-frequency detail recovery. Extensive experimentation validates the framework's broad generalizability across challenging ultra-dark scenarios, outperforming current state-of-the-art methods across 4 real and synthetic benchmark datasets over 8 metrics (e.g., **5.8%**↑ in PSNR, **26.6%**↑ in LPIPS, and **13.9%**↑ in RMSE).

## CCS CONCEPTS

• **Computing methodologies** → *Scene understanding*.

## KEYWORDS

Nighttime vision, super-resolution, coupled degradations, bi-level

## 1 INTRODUCTION

Enhancing visibility and enlarging the resolution of ultra-dark images simultaneously is a daunting task with substantial real-world significance for fields such as intelligent surveillance and nocturnal

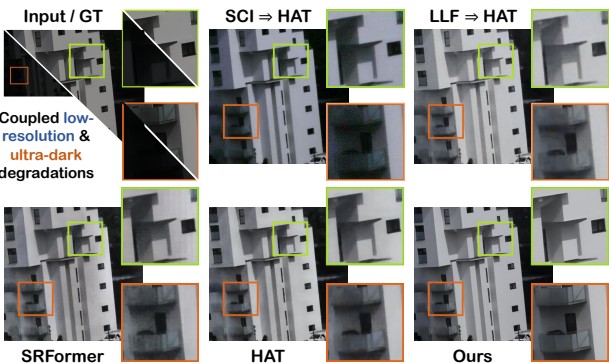

**Figure 1: Visual comparison of advanced LLE [28, 37] and SR methods [3, 53] applied both independently and in a cascaded manner to inputs with coupled degradations.**

autonomous driving [19, 27, 36, 38]. Due to inherent limitations in imaging devices and constraints posed by environmental lighting conditions, captured data frequently exhibits coupled degradations characterized by low resolution and extreme darkness. Imaging devices may struggle to capture clear details in poorly lit environments, resulting in images of low resolution; concurrently, insufficient ambient lighting exacerbates the darkness of the images, making content difficult to discern. This paper addresses the integrated challenge of enhancing brightness and increasing resolution in ultra-dark images plagued by these intertwined degradations.

Capturing images in ultra-low-light settings introduces a plethora of challenges that amplify the complexity of this joint task, including *uneven exposure resulting in highly irregular lighting, diminished contrast, color inaccuracies, and an overflow of artifacts.* Standard image Super-Resolution (SR) techniques [11, 21, 39, 41], crafted with modular techniques for image resolution enhancement under normal-lighting scenarios, cannot be straightforwardly adapted to enhance the luminance and resolution of images captured in low-light conditions. Indeed, the direct application of these techniques would inevitably amplify hidden noise, blur, and artifacts present in darkness, leading to unnatural edges and textures and deviating from the primary goal of super-resolution. Contrarily, recent Low-Light Enhancement (LLE) methods [19, 26, 43], while capable of brightening, fall short in concurrently amplifying resolution and authentically enhancing high-frequency details. This prompts a further inquiry: *Can LLE and SR be effectively combined in a simple "A+B" cascaded format to achieve the desired outcomes?* Upon evaluation, we ascertain that this direct "A+B" does not address the entangled degradation factors at the data level, possibly akin to a "A×B" degraded form, with ongoing shortcomings in enhancing brightness and rendering texture details.

As illustrated in Fig. 1, we present a visual comparison of two cutting-edge LLE methods – SCI [28] and LLFormer [37] – alongside two normal-light SR techniques, HAT [3] and SRFormer [53].

A closer examination reveals that employing HAT and SRFormer independently falls short in recapturing fine details, producing blurred artifacts. Similarly, cascaded LLE⇒SR approaches (e.g., SCI⇒HAT, LLFormer⇒HAT) also fail to restore adequate brightness, exacerbating noise and structural distortions when magnified. In stark contrast, our proposed method generates natural and authentic exposure and color fidelity, alongside improved structural clarity and texture detail. In addition to the methods previously mentioned, a few recent studies have emerged focusing on super-resolution within low-light scenes [5, 10]. They tend to rely on simple brightness corrections and resolution scaling on synthetic datasets, leading to poor generalization in real-world scenarios. Thus, we summarize the two primary shortcomings limiting the efficacy of existing methods: (i) *The failure to recognize the intricacies of degradation-coupled data, which extends beyond a simplistic additive enhancement model. This overlooks the intrinsic bidirectional cooperation necessary for joint processing.* (ii) *A heavy reliance on empirical network design and manual aggregation of losses. Such methods disregard the guiding principles of physical image formation, and overlook the crucial role that suitably chosen loss constraints play in facilitating cooperative learning between intertwined tasks.*

Stemming from these insights, this paper seeks to explore a tri-level optimization perspective that formulates the cooperative relationships and devises a corresponding solution strategy. We propose TriCo, aiming to automate the optimization of these weighted constraints with hyper-variable and the coupling dependencies between two entangled tasks, striving to achieve a unified enhancement. Specifically, we initiate the process with an illumination interpolation mapping inspired by Retinex theory, yielding a brightened reflectance that serves as the foundation for subsequent feature-level super-resolution enhancement. We leverage universal foundational semantic model priors and illumination features under the self-regularized luminance constraint to provide dual guidance for the super-resolution process, specifically targeting the compensation of high-frequency details. On the algorithmic front, we have crafted a phased gradient-response algorithm as our training mechanism, meticulously designed to synergize the optimization of three key variables while offering dynamic gradient feedback throughout the training phase, thereby ensuring streamlined training efficiency and rapid convergence. In summary, our contributions are fourfold:

- We propose **TriCo**, the first to introduce a **Tri**-level optimization perspective that explicitly models the bidirectional **Co**operative relationship of illumination learning and super-resolution, formulating a solution to synergistically brighten and enlarge images afflicted with coupled low-resolution and ultra-dark degradations.
- We establish an Upper (U)-Middle (M)-Lower (L) level nested formulation, which in its M-L level, explicitly delineates the collaborative dependency of two entangled tasks. In the U-M level, we integrate hyper-variables to autonomously enforce positive constraint feedback, thus dismantling the reliance on manual trial-and-error intervention.
- We propose a Phased Gradient-Response (PGR) algorithm as the training mechanism, designed to synergistically optimize three variables while providing dynamic gradient feedback, thus achieving efficient training and rapid convergence.

- We propose an Integrated Hybrid Expert Modulator (IHEM) that acts as a conduit between illumination prior cues (i.e., intrinsic attributes) and generic semantic model features (i.e., SAM), facilitating an adaptive pixel-level guidance for the restoration of high-frequency details.

Extensive experimentation validates the framework's broad generalizability and performance advantages across 4 real and synthetic benchmark datasets over 8 metrics (e.g., **5.8%**↑ in PSNR, **26.6%**↑ in LPIPS, and **13.9%**↑ in RMSE).

## 2 RELATED WORK

**Enhancing Low-Light Images.** LLE's goal is to make images engulfed in darkness visible. Early works generally concentrated on leveraging handcrafted priors and empirical insights for LLE, such as Retinex model [13, 17] for separate treatment of illumination and reflection. Recent advancements have been seen with models based on convolutional neural networks, addressing these fundamental challenges [9, 26, 28]. Typically, such techniques always rely on manually selecting complex losses. Instead, we introduce the tri-level automated strategy to pinpoint beneficial constraint feedback, diverging from the reliance on empirical hyperparameter tuning.

**Normal/Low-light Image Super-Resolution.** Normal-light SR task generates high resolution images from low resolution inputs under standard lighting conditions. Recently, a large number of methods based on convolutional neural networks have emerged to continuously refresh the performance [11, 21, 42]. With the growing popularity of transformer-based technologies, many leading-edge methods [3, 4, 22, 53] have been developed for super-resolution enhancement, including SwinIR [22], Restormer [46], the recently proposed SRFormer [53] and HAT [3]. Drawing on this, we fuse semantic cues from universal models [15, 48] with illumination attributes for modulation, meticulously steering the detail restoration of reflectance features and maintaining color consistency.

Moreover, recent forays into super-resolution focused on low-light imagery, have yet to yield satisfactory results [5, 10, 32]. For instance, Cheng *et al.* [5] proposed a light-guided and cross-fusion U-Net, featuring an intensity estimation unit, targets uneven-light image super-resolution. Yet, its sole reliance on pixel shuffling for resolution enlargement introduces notable color distortion and a lack of clarity in structure. A potential reason is that previous approaches did not account for the coupled collaboration between the two tasks, treating them in isolation. Hence, we employ hierarchical optimization to model and solve this multi-tiered coupled task.

**Bi-level Optimization.** Bi-level Optimization is the hierarchical mathematical program, where the feasible region of upper-level task is restricted by the solution set mapping of lower-level task and the two task are mutually reinforced [16]. Subsequently, the bi-level optimization framework has been investigated in view of many important applications in the fields of machine learning and computer vision e.g., hyper-parameter optimization [12, 29], multi-task and meta learning [29, 35], neural architecture search [45, 54], and image processing and analysis [24, 25]. Motivated by the above observations, We explicitly consider the collaborative relationship between super-resolution and brightness adjustment tasks, constructing a novel perspective with tri-level optimization for modeling and solving.

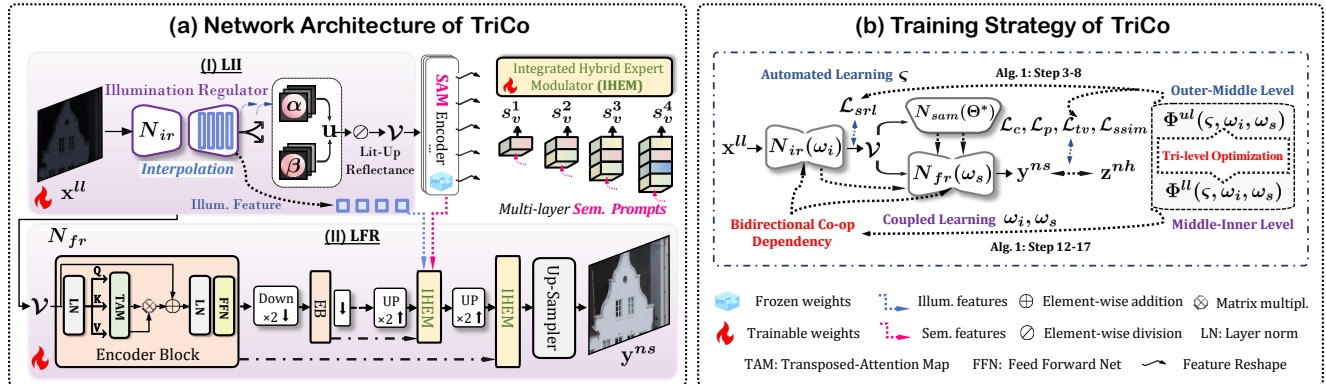

Figure 2: The overall TriCo framework. (a) initiates with an interpolation-based illumination regulator $N_{ir}$ (parameterized by $\omega_i$), producing a lit-up reflectance $v$. Then $v$ feeds into a frozen SAM for multi-scale semantic prompts, while concurrently undergoing refinement for the LFR ($N_{fr}$ parameterized by $\omega_s$) with guidance from IHEM. (b) establishes tri-level learning paradigm with a phased gradient-response algorithm to foster a collaborative, automated, and efficient training process.

## 3  METHODOLOGY

Our TriCo aims to transform extremely dim, low-resolution images $x^{ll}$ into luminance-friendly, super-resolution counterparts $y^{ns}$. In the following, we firstly delve into the architecture of our model, followed by the details of the proposed learning strategy.

### 3.1  Illumination-Guided Integrated Network

Acknowledging the consensus that direct enlarging of dark images can result in the loss of details, noise amplification, and artifacts, we meticulously crafted an illumination-guided integrated network. This network does not merely cascade brightness adjustment with resolution enhancement modules; instead, it adopts a more profound integration approach to ensure a holistic improvement. As depicted in Fig. 2, our network adopts a top-down integration approach, seamlessly incorporating an illumination adjustment sub-network for initial brightness enhancement and a feature-level refinement sub-network for resolution upscaling. A pivotal bridging component (i.e., Integrated Hybrid Expert Modulator, IHEM) is introduced within this structure, leveraging illumination and semantic priors as cues for guided refinement. This architecture is summarized into two phases: (i) the Learning Interpolated Illuminance (LII), focusing on adjusting the initial luminance, and (ii) the Learning Feature Refinement (LFR) for super-resolution, dedicated to enhancing and enlarging the image details at the feature level.

**From LII to LFR.** LII performs the initial mapping from "low-light" to "normal-light". Based on Retinex theory, the normalized illumination map satisfies the inequality within the dynamic range: $0 \le x^{ll} \le u \le I$. Thus, LII is designed to construct an interpolation mapping to estimate the illumination map $u$. Finally, the initial reflectance map $v$ is obtained by applying element-wise division to $u$. The initial reflectance map $v$ is then input to the LFR sub-network (i.e., $N_{fr}$) for fine-grained feature modulation, ensuring that the upsampling process generates more high-frequency details:

$$\begin{cases} u = \alpha \cdot x^{ll} + \beta \cdot I, \ \{\alpha, \beta\} = N_{ir}(\omega_i; x^{ll}), \\ v = x^{ll} \oslash u, \ y^{ns} = N_{fr}(\omega_s; v), \end{cases} \quad (1)$$

where the interpolation factors $\alpha$ and $\beta$ are generated by the underlying Unet-style illumination regulator $N_{ir}$ and satisfy the constraints within the unit interval, with their sum equaling 1. $\omega_i$ and $\omega_s$ are network parameters of $N_{ir}$ and $N_{fr}$, respectively. Finally, we introduce a dynamic grid up-sampling module [7] to enlarge the image dimensions. For specific architectural details of $N_{ir}$ and $N_{fr}$, please refer to the *Supplementary Material*. The uniqueness of LII lies in its reliance solely on a single luminance loss[1] for unsupervised learning, eliminating cumbersome training with multiple stages and losses.

Also, we feed $v$ into a pre-trained large-scale base semantic model (i.e., SAM [15, 48]) to extract multi-layer semantic features: $N_{sam}[v|\Theta^*_{sam}] = [f_s^{[1]}, \cdots, f_s^{[o]}, \cdots, f_s^{[J]}], o = 0, \cdots, J$. We note that the multi-scale illuminance features and semantic features can serve as expert cues containing degradation priors (i.e., exposure and color information of different local areas). Therefore, we design the Integrated Hybrid Expert Modulator (IHEM) to modulate reflectance features layer-by-layer within the LFR sub-network's decoder, guiding the generation of high-frequency texture details.

**IHEM.** As illustrated in Fig. 3, the structural details of the IHEM are showcased. Denote each layer's reflectance feature in the decoder as $f_r^{[o]}$. First, $f_s^{[o]}$ and $f_r^{[o]}$ undergo layer normalization, 1×1 convolution, and 3×3 depth-wise convolution, leading to the formation of *semantic query* ($\tilde{Q}_s \in \mathbb{R}^{\tilde{H}\tilde{W}\times\tilde{C}}$), *reflection key* ($\tilde{K}_r \in \mathbb{R}^{\tilde{C}\times\tilde{H}\tilde{W}}$), and *reflection value* ($\tilde{V}_r \in \mathbb{R}^{\tilde{H}\tilde{W}\times\tilde{C}}$) projections. Following this, we derive the **S**emantic-**In**duced **Att**ention (**S-InA**) map, $A_{S-InA} \in \mathbb{R}^{\tilde{C}\times\tilde{C}}$, which is normalized via Softmax. The reflectance feature $f_r^{[o]}$ is subsequently updated via the transposed **S**emantic-**In**duced **R**esponse (**S-InR**, $W_{S-InR}$):

$$W_{S-InR} = \text{Conv}[\tilde{V}_r \otimes \overbrace{\text{Softmax}((\tilde{Q}_s \otimes \tilde{K}_r)/\tau_1)}^{A_{S-InA}}] + f_r^{[o]}, \quad (2)$$

where $\tau_1$ represents a learnable scaling factor that adjusts the magnitude of the product of $\tilde{K}_r$ and $\tilde{Q}_s$. $\otimes$ denotes the element-wise

---

[1]Please refer to the self-regularized luminance loss in Eq. (8).

multiplication. Then, a feed-forward network *FFN* [14, 46] is employed to facilitate improved content reconstruction, denoted as: $\tilde{f}_r^{[o]} = FFN(W_{\text{S-InR}})$. For architectural specifics of *FFN*, please consult the *Supplementary Material*.

Similarly, we output multi-layer illumination features $f_i^{[o]}$ from the LII decoder, which, along with the semantically guided features $\tilde{f}_r^{[o]}$, jointly undergo the **Illumination-Induced Att**ention (**I-InA**, $A_{\text{I-InA}} \in \mathbb{R}^{\tilde{C} \times \tilde{C}}$) process. Initially, each is processed via layer normalization, 1×1 convolution, and 3×3 depth-wise convolution, leading to the formation of illumination query ($\bar{Q}_i \in \mathbb{R}^{\tilde{H}\tilde{W} \times \tilde{C}}$), reflection key ($\bar{K}_r \in \mathbb{R}^{\tilde{C} \times \tilde{H}\tilde{W}}$), and reflection value ($\bar{V}_r \in \mathbb{R}^{\tilde{H}\tilde{W} \times \tilde{C}}$) projections. Then $\tilde{f}_r^{[o]}$ undergoes dynamic enhancement via the transposed **Illumination-Ind**uced **R**esponse (**I-InR**, $W_{\text{I-InR}}$):

$$W_{\text{I-InR}} = \text{Conv}[\Phi_R \otimes \overbrace{\text{Softmax}((\bar{Q}_i \otimes \bar{K}_r)/\tau_2)}^{A_{\text{I-InA}}}] + \tilde{f}_r^{[o]}, \quad (3)$$

where $\tau_2$ is a learnable scaling factor. Subsequently, $W_{\text{I-InR}}$ is processed by *FFN*, yielding the doubly modulated reflection feature $\tilde{f}r^{[o]} = FFN(W_{\text{I-InR}})$.

## 3.2 Tri-level Optimization Formulation

**Bidirectional Co-op Dependency.** Existing methods often focus on a single task, either brightness adjustment or resolution enhancement, seldom considering the interdependent coupling between the two. Recognizing that LII and LFR, can mutually reinforce each other—where precise luminance improvement by LII can facilitate better super-resolution outcomes in LFR, and conversely, the detailed enhancement by LFR can enhance the illumination learning in LII—we model these consecutive learning tasks as a hierarchical optimization problem, formalized as follows:

$$\begin{cases} \min_{\omega_i} \Phi^{ul}(\omega_i, \omega_s^*; \{\mathcal{D}_{ul}\}), \ s.t., \ \omega_s^* \in \mathcal{P}_l(\omega_i), \\ \mathcal{P}_l(\omega_i) := \arg\min_{\omega_s} \Phi^{ll}(\omega_i, \omega_s; \{\mathcal{D}_{ll}\}), \end{cases} \quad (4)$$

where $\mathcal{P}_l(\cdot)$ denotes the solution set, with $\mathcal{D}_{ll}$ and $\mathcal{D}_{ul}$ representing the lower and upperlevel datasets, respectively.

The hierarchical formulation explicitly delineates the collaborative training modality between $N_{ir}$ and $N_{fr}$. This collaboration is heavily contingent upon the judicious selection of lower and upper level objectives, ensuring that the sub-networks can reciprocally foster enhancement and positive feedback. This raises a pivotal question: *How can we automate the assignment of high hyperparameters that significantly foster positive influences on the learning tasks?* Delving deeper into this inquiry, we transcend the confines of the hierarchical optimization framework and venture into an expanded horizon—establishing a nested optimization problem that encompasses both lower and upper levels, aimed at the autonomous learning of beneficial constraints for two learning tasks.

**Tri-level Constraint Modeling.** Evidently, to automate the determination of constraints that significantly influence the learning tasks through weight allocation, we introduce a novel concept, the hyper-variable $\varsigma$. This hyper-variable, along with the two preceding variables $\omega_i$ and $\omega_s$, forms a new set of constraint relationships, thereby constituting a hierarchical optimization problem based on

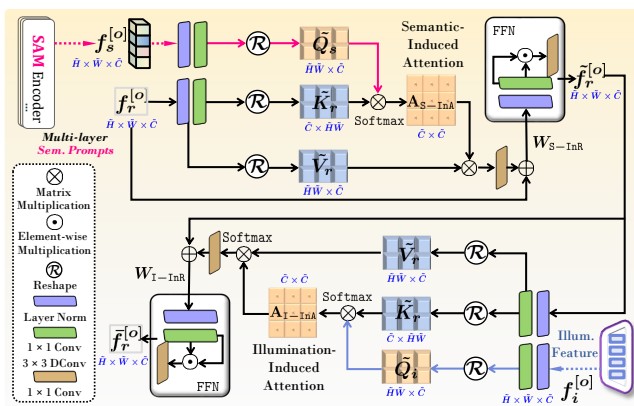

**Figure 3: The detailed architecture of IHEM.**

three variables, as shown below:

$$\begin{cases} \min_{\varsigma} \Phi^{ul}(\varsigma, \omega_i^*, \omega_s^*; \{\mathcal{D}_{ul}\}), \ s.t., \ (\omega_i^*, \omega_s^*) \in \mathcal{P}_l(\varsigma), \\ \mathcal{P}_l(\varsigma) := \arg\min_{\omega_i, \omega_s} \Phi^{ll}(\varsigma, \omega_i, \omega_s; \{\mathcal{D}_{ll}\}), \end{cases} \quad (5)$$

where $\omega_i^*$ and $\omega_s^*$ represent the best-reponse for a given $\varsigma$. These three variables are highly interdependent and dynamically influence each other throughout the training process. This modeling approach offers significant advantages: firstly, it explicitly defines the mathematical relationships between multiple variables, allowing for dynamic feedback during training, thereby enhancing training efficiency. Secondly, it automates the determination of constraints' positive feedback, overcoming the reliance on manual hyper-parameter tuning based on empirical knowledge, thereby reducing the need for extensive manual intervention.

## 3.3 Algorithmic Procedure

Moving forward, we devise a Phased Gradient-Response (PGR) algorithm that iterates from the upper to the lower layers, serving as the training strategy. Specifically, we define a comprehensive function as the weighted sum of multiple losses related to the hypervariables $\varsigma$, addressing various specific attributes (e.g., brightness, color, exposure, smoothness, and content) pertinent to multi degradation restoration tasks. The total loss is defined as follows:

$$\mathcal{L}_{total}(\varsigma, \omega_i, \omega_s; \{\mathcal{D}\}) = \sum_{u=1}^{N} \varsigma_u \cdot \mathcal{L}_u(\varsigma, \omega_i, \omega_s), \ \mathcal{L}_u \in \mathcal{T}, \quad (6)$$

where the hyper-variable is denoted as $\varsigma := \{\varsigma_u\}_{u=1}^N \in \mathbb{R}^N$. $\mathcal{T}$ represents the loss selection set. Please refer to Sec. 3.4 for $\mathcal{T}$. To prevent ambiguous solutions during training and safeguard against overfitting, we introduce a regularization constraint term for $\varsigma$, thereby reformulating the total loss as: $\mathcal{L}_{total}(\varsigma, \omega_i, \omega_s; \{\mathcal{D}\}) = \sum_{u=1}^{N} \frac{1}{2\varsigma_u} \cdot \mathcal{L}_u(\varsigma, \omega_i, \omega_s) + \ln(1 + \varsigma_u^2)$. The training set is divided into proportions denoted by $\eta$, thus the upper and lower levels are abstractly defined as: $\Phi^{ul} := \mathcal{L}_{total}(\varsigma, \omega_i, \omega_s; \mathcal{D}_{ul})$, $\Phi^{ll} := \mathcal{L}_{total}(\varsigma, \omega_i, \omega_s; \mathcal{D}_{ll})$. Next, we decompose the problem into two stages of hierarchical optimization to solve the tri-level coupled problem step by step.

---

**Algorithm 1** Optimization strategy for TriCo.

---

**Require:** Initialize $\omega := \{\omega_i, \omega_s\}$, with $\varsigma$ as a unit vector. Learning rate: $\gamma_u, \gamma_l, o_u$ and $o_l$; Total iterations $\mathcal{K}$. Split $\{\mathcal{D}\} := \{\mathcal{D}_{ul}\} \cup \{\mathcal{D}_{ll}\}$ with partition ratio $s$. Set candidate loss space $\mathcal{T}$.

**Ensure:** The optimal parameters $\varsigma, \omega$.

1: % % S1: Automated learning for $\varsigma$.
2: **while** not converged **do**
3:      % % *Upper-level variable probe*:
4:      $\hat{\omega} \leftarrow \omega - \gamma_l \frac{\partial \Phi^{ll}(\varsigma, \omega)}{\partial \omega}, \omega^{\pm} \leftarrow \omega \pm \lambda \frac{\partial \Phi^{ll}(\varsigma, \hat{\omega})}{\partial \hat{\omega}}$
5:      $\mathcal{A}_{\omega} \leftarrow \frac{1}{2\lambda}\left(\frac{\partial \Phi^{ll}(\varsigma, \omega^+)}{\partial \varsigma} - \frac{\partial \Phi^{ll}(\varsigma, \omega^-)}{\partial \varsigma}\right)$
6:      $\varsigma \leftarrow \varsigma - \gamma_u \frac{\partial \Phi^{ul}(\varsigma, \hat{\omega})}{\partial \varsigma} + \gamma_l \mathcal{A}_{\omega}$
7:      % % *Middle-level variable probe*:
8:      $\omega \leftarrow \omega - \gamma_l \frac{\partial \Phi^{ll}(\varsigma, \omega)}{\partial \omega}$
9: **end while**
10: % % S2: Optimization for $\{\omega_i, \omega_s\}$ with frozen $\varsigma$.
11: **while** not converged **do**
12:      % % *Middle-level variable probe*:
13:      $\hat{\omega}_s \leftarrow \omega_s - o_l \frac{\partial \Phi^{ll}(\omega_i, \omega_s)}{\partial \omega_s}, \omega^{\pm} \leftarrow \omega_s \pm \lambda \frac{\partial \Phi^{ll}(\omega_i, \hat{\omega}_s)}{\partial \omega_s}$
14:      $\mathcal{B}_{\omega_s} \leftarrow \frac{1}{2\lambda}\left(\frac{\partial \Phi^{ll}(\omega_i, \omega_s^+)}{\partial \omega_i} - \frac{\partial \Phi^{ll}(\omega_i, \omega_s^-)}{\partial \omega_i}\right)$
15:      $\omega_i \leftarrow \omega_i - o_u \frac{\partial \Phi^{ul}(\omega_i, \hat{\omega})}{\partial \omega_i} + o_l \mathcal{B}_{\omega_s}$
16:      % % *lower-level variable probe*:
17:      $\omega_s \leftarrow \omega_s - o_l \frac{\partial \Phi^{ll}(\omega_i, \omega_s)}{\partial \omega_s}$
18: **end while**

---

**Gradient-Response Algorithm.** Following the first-order gradient algorithm based on hierarchical optimization [24], we compute the composite upper gradients based on the best-response from the lower optimization. We first calculate the upper-level gradient:

$$\nabla_{\varsigma} \Phi^{ul}(\varsigma, \omega) = \frac{\partial \Phi^{ul}(\varsigma, \omega^*(\varsigma))}{\partial \varsigma} + \frac{\partial \Phi^{ul}(\varsigma, \omega^*(\varsigma))}{\partial \omega} \nabla_{\varsigma} \omega^*(\varsigma). \quad (7)$$

For simplicity, we define the lower-level variables as $\omega := \{\omega_i, \omega_s\}$. The second term, the coupled gradient, is denoted as $\mathcal{A}_{\omega}$. Subsequently, based on a single-step gradient descent to approximate the best-response, we calculate the finite difference approximation [23] for the coupled gradient $\mathcal{A}_{\omega}$ as $\mathcal{A}_{\omega} = \frac{1}{2\lambda}\left(\frac{\partial \Phi^{ll}(\varsigma, \omega^+)}{\partial \varsigma} - \frac{\partial \Phi^{ll}(\varsigma, \omega^-)}{\partial \varsigma}\right)$, where $\omega^{\pm} \leftarrow \omega \pm \lambda \frac{\partial \Phi^{ll}(\varsigma, \omega)}{\partial \omega}$, and $\lambda$ denotes a constant learning rate. For the second phase, a similar derivation to the first phase is implemented. Given the optimal hyper-variable $\varsigma^*$ obtained from the first stage, we compute the upper-level gradient with respect to the variable $\omega_i$: $\nabla_{\omega_i} \Phi^{ul}(\omega_i, \omega_s) = \frac{\partial \Phi^{ul}_{\varsigma^*}(\omega_i, \omega_s^*(\omega_i))}{\partial \omega_i} + \mathcal{B}_{\omega_s}$, where $\mathcal{B}_{\omega_s} = \frac{\partial \Phi^{ul}_{\varsigma^*}(\omega_i, \omega_s^*(\omega_i))}{\partial \omega_s} \nabla_{\omega_i} \omega_s^*(\omega_i)$. Ultimately, the optimization process across both stages is amalgamated to form our training strategy, which is summarized in Alg. 1.

## 3.4 Loss Candidate Space

As illustrated in Fig. 2, we propose a set of five specific loss objectives constituting a candidate space that encapsulates the model's constraints on brightness, color, exposure, smoothness, and content attributes, denoted as the set $\mathcal{T}$, as follows:

- Self-regularized luminance loss: To ensure that the generated reflectance aligns with the luminance attributes of large-scale natural ImageNet dataset [8] in a consistent distribution, we design $\mathcal{L}_{srl}$:

$$\mathcal{L}_{srl}(\mathbf{v}) = e^{|\bar{\mathbf{v}}_c - \mu_c - \sigma_c|} - 1, \ c \in \{R, G, B\}, \quad (8)$$

where $\bar{\mathbf{v}}_c$ signifies the operation of computing the mean across channels. Channel means and standard deviations are $\mu_c = [0.485, 0.456, 0.406]$ and $\sigma_c = [0.229, 0.224, 0.225]$.

- Content reconstruction loss: We employ the standard reconstruction loss between $\mathbf{y}^{ns}$ and $\mathbf{z}^{nh}$ utilizing the $L_1$ norm:

$$\mathcal{L}_c(\mathbf{y}^{ns}, \mathbf{z}^{nh}) = \frac{1}{hwc} \sum_{i,j,k} |\mathbf{y}^{ns}_{i,j,k} - \mathbf{z}^{nh}_{i,j,k}|, \quad (9)$$

where $h, w, c$ are the height, width, and channel count.

- Semantic perceptual loss: We utilize a perceptual loss function to maintain semantic congruence between $\mathbf{y}^{ns}$ and $\mathbf{z}^{nh}$:

$$\mathcal{L}_p(\mathbf{y}^{ns}, \mathbf{z}^{nh}) = ||\text{VGG19}_j(\mathbf{y}^{ns}) - \text{VGG19}_j(\mathbf{z}^{nh})||_1, \quad (10)$$

where $j$ indicates the j-th feature extraction layer, which includes layers from $\text{conv1}, \cdots, \text{conv5}$.

- Structural similarity loss: We employ the SSIM loss $\mathcal{L}_{ssim}$ to maintain the structural similarity between $\mathbf{y}^{ns}$ and $\mathbf{y}^{nh}$.

- Smoothness loss: We incorporate a total variation metric [33] to reduce noise and enhance image smoothness:

$$\mathcal{L}_{tv}(\mathbf{y}^{ns}) = \sum_{\xi \in \pi} (|\nabla_h \mathbf{y}^{ns}_{\xi}| + |\nabla_v \mathbf{y}^{ns}_{\xi}|), \quad (11)$$

where $\pi = \{R, G, B\}$, $\nabla_h$ and $\nabla_v$ are the horizontal and vertical gradient operators, respectively.

# 4 EXPERIMENTS

## 4.1 Experimental Settings

**Datasets and Metrics.** We evaluated the benchmark performance of all compared methods across four datasets: 1) RELLISUR [1][2], 2) DarkFace [44][3], 3) Dark-Zurich [34], and 4) Cityscapes [6]. Due to space constraints, please refer to the *Supplementary Material* for details on the data preparation of four datasets. In the evaluation phase, we employ five full-reference metrics to assess the performance, namely PSNR [2], SSIM [40], Learned Perceptual Image Patch Similarity (LPIPS) [50], Root Mean Square Error (RMSE), Feature-based Similarity Index (FSIM) [49]. Additionally, we introduce three no-reference assessments, namely Natural Image Quality Evaluator (NIQE) [31], Blind/Referenceless Image Spatial Quality Evaluator (BRISQUE) [30] and MetaIQA [55], to evaluate non-paired metrics.

**Implementation Details.** We adhere to the tri-level learning strategy as outlined in Alg. 1 for our network training, with the total number of iterations set to 150,000. We utilize the Adam optimizer with beta values configured at $[0.9, 0.999]$. The initial learning rates for the upper and lower layers of the two stages are set to $\gamma_u = 1e-4$, $\gamma_l = 2e-4$, $o_u = 1e-4$ and $o_l = 1e-4$, respectively. A cosine annealing restart strategy is implemented for cyclic learning rate scheduling. The dataset $\{\mathcal{D}\}$ is partitioned into $\{\mathcal{D}_{ul}\} \cup \{\mathcal{D}_{ll}\}$ and at a distribution ratio of $1 : 5$. Experiments are conducted

---

[2] https://vap.aau.dk/rellisur/
[3] https://flyywh.github.io/CVPRW2019LowLight/

**Table 1: Quantitative comparison on _RELLISUR_ dataset for _@×2_ and _@×4_ tasks. The best three results are bolded in red, green, and blue, indicating first, second, and third places, respectively.**

| Method | Public. | RELLISUR @ ×2 | | | | | RELLISUR @ ×4 | | | | |
|---|---|---|---|---|---|---|---|---|---|---|---|
| | | ❶PSNR↑ | ❷SSIM↑ | ❸LPIPS↓ | ❹RSME↓ | ❺FSIM↑ | ❶PSNR↑ | ❷SSIM↑ | ❸LPIPS↓ | ❹RMSE↓ | ❺FSIM↑ |
| SRResNet | CVPR'17 | 18.153 | 0.667 | 0.451 | 0.128 | 0.483 | 17.597 | 0.684 | 0.581 | 0.137 | 0.410 |
| RDN | CVPR'18 | 18.794 | 0.701 | 0.455 | 0.120 | 0.501 | 18.219 | 0.701 | 0.584 | 0.128 | 0.428 |
| SRFBN | CVPR'19 | 18.427 | 0.662 | 0.510 | 0.125 | 0.476 | 17.676 | 0.665 | 0.640 | 0.136 | 0.407 |
| PAN | ECCV'20 | 18.789 | 0.693 | 0.450 | 0.119 | 0.491 | 18.106 | 0.700 | 0.559 | 0.129 | 0.427 |
| MIRNet | ECCV'20 | 21.052 | 0.720 | 0.436 | 0.095 | 0.501 | 19.784 | 0.704 | 0.599 | 0.109 | 0.419 |
| SwinIR | ICCV'21 | 18.383 | 0.640 | 0.577 | 0.125 | 0.464 | 17.531 | 0.663 | 0.688 | 0.139 | 0.418 |
| Restormer | CVPR'22 | 21.217 | 0.727 | 0.385 | 0.095 | 0.505 | 20.290 | 0.720 | 0.492 | 0.106 | 0.425 |
| LCUN | TCSVT'22 | 18.911 | 0.684 | 0.531 | 0.131 | 0.476 | 18.463 | 0.657 | 0.644 | 0.131 | 0.370 |
| SRFormer | CVPR'23 | 19.554 | 0.704 | 0.469 | 0.110 | 0.492 | 18.792 | 0.705 | 0.613 | 0.121 | 0.430 |
| HAT | CVPR'23 | 20.213 | 0.719 | 0.454 | 0.103 | 0.501 | 19.751 | 0.715 | 0.561 | 0.110 | 0.421 |
| †ZeroDCE⇒‡HAT | | 12.927 | 0.354 | 0.698 | 0.194 | 0.412 | 12.524 | 0.321 | 0.739 | 0.197 | 0.362 |
| †SCI⇒‡HAT | | 14.963 | 0.439 | 0.591 | 0.200 | 0.405 | 14.776 | 0.452 | 0.697 | 0.205 | 0.362 |
| ‡LLFormer⇒†HAT | | 21.218 | 0.720 | 0.455 | 0.093 | 0.499 | 20.135 | 0.718 | 0.575 | 0.105 | 0.429 |
| Ours | - | 22.456 | 0.744 | 0.304 | 0.080 | 0.508 | 21.056 | 0.731 | 0.432 | 0.006 | 0.429 |

† signifies training using the ×1 low-light RELLISUR dataset for LLIE. ‡ indicates training using the ×2 or ×4 for normal-light SR.

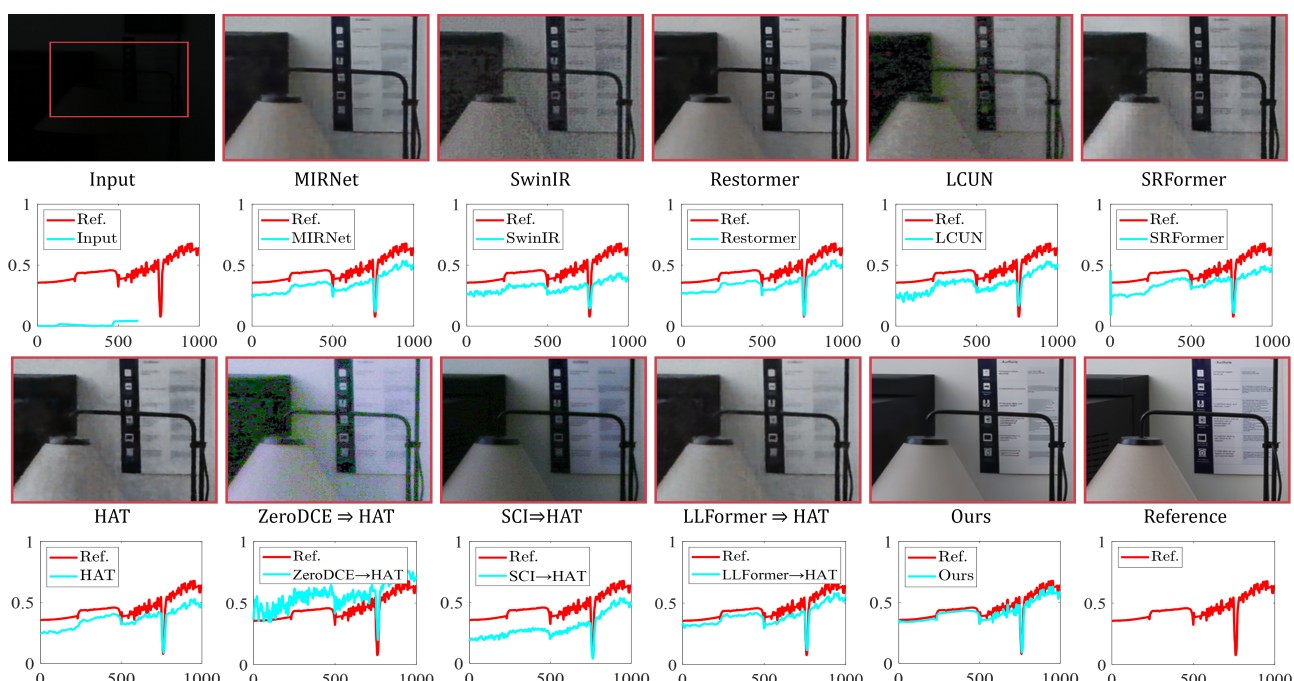

**Figure 4: Visual assessments using _RELLISUR_ examples for a ×2 magnification task. The signal plots depict variations in pixel intensities between the produced images and the benchmark image, traced across arbitrarily chosen line segments.**

using PyTorch version 2.0.1, which supports CUDA 11.7, on a single NVIDIA RTX A6000 GPU with 48GB of RAM.

**Compared Methods.** To substantiate the efficacy of our proposed methodology, we conduct a comprehensive comparison with a diverse array of SOTA methods in LLE and SR. Specifically, we meticulously benchmark against 3 emblematic LLE techniques, namely ZeroDCE [20], SCI [28], and LLFormer [37], alongside 10 SR algorithms, which include 9 under normal lighting conditions—SRResNet [18], RDN [51], SRFBN [21], PAN [52], SwinIR [22], MIRNet [47], Restormer [46], SRFormer [53], and HAT [3]—and

one dedicated to low-light scenarios, LCUN [5]. Notably, the enhancement results on the RELLISUR dataset for the sole low-light SR method, LCUN, were furnished by the authors themselves. To ensure a fair comparison, we retrain the publicly available codes of all competing methods on the training set of the RELLISUR dataset. We opt for HAT as the subsequent magnification model, cascading it with three distinct LLE methods. It is noteworthy that, given ZeroDCE and SCI operate in an unsupervised manner, we train them on the ×1 low-light RELLISUR dataset for the initial "brightening"

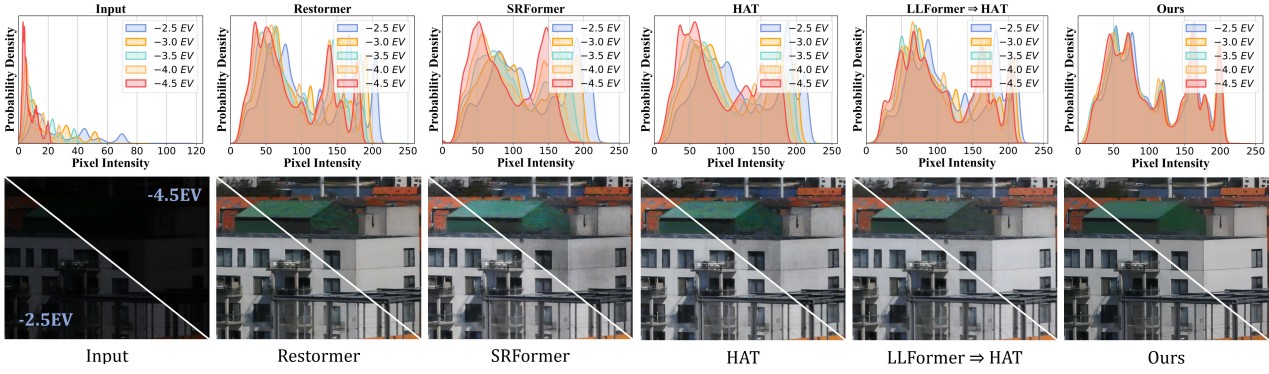

**Figure 5: Illustrating (*Above*) the probability density histogram trends and (*Below*) enhancement outcomes for the same sample (i.e., 00018) under five different levels of darkness. Note that from -2.5EV to -4.5EV indicates increasing darkness.**

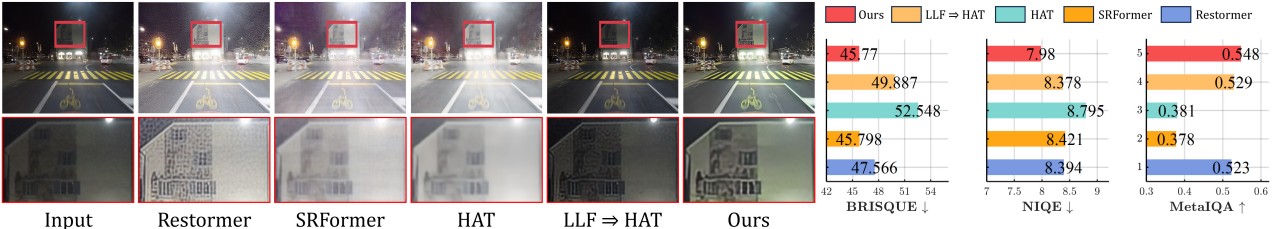

**Figure 6: Visual comparisons (*Left*) and quantitative results on three metrics (*Right*) for enhancing brightness and enlarging low-light images on real nighttime Dark-Zurich samples. Note here, LLFormer⇒HAT is abbreviated as LLF⇒HAT.**

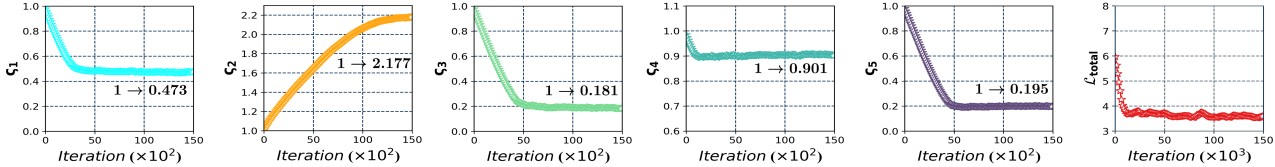

**Figure 7: Illustrating the convergence curve of the hyper-variable $\{\varsigma_u\}_{u=1}^{5}$ and the total loss $\mathcal{L}_{total}$, based on Alg. 1.**

task. Moreover, considering HAT serves as a post-processing module for super-resolving the enhanced images, it is trained on the ×2 and ×4 normal-light RELLISUR datasets, designated as ‡HAT, to differentiate from the original HAT configuration.

### 4.2 Algorithmic Mechanism Evaluation

Following Alg. 1, we undertake a tri-level automated training regimen to sequentially optimize the three variables. Fig. 7 illustrates the convergence trend of the outermost variable and the overall loss function throughout the iterations. During the S1 cycle in Alg. 1 (refer to steps 1 to 8), the upper-level variable $\{\varsigma_u\}_{u=1}^{5}$ evolves from an initial unit vector to eventually converge to [0.473, 2.177, 0.181, 0.901, 0.195]. This convergence elucidates that the constraints positively influencing the learning task, as hypothesized by our algorithm, are indeed effective. It is possible to autonomously identify which constraints significantly foster a positive impetus for the learning task through adaptive weight allocation. Notably, the top three constraints—perceptual constraint, smoothness constraint, and reconstruction constraint—play a pivotal role in augmenting performance, underscoring the efficacy of our proposed approach in leveraging these constraints for enhanced learning outcomes.

### 4.3 Comparisons with State-of-the-Art

**Evaluation on RELLISUR.** Tab. 1 presents the quantitative results for low-light super-resolution tasks at ×2 and ×4 scales on the *RELLISUR* dataset. While cascading strategies prove effective, the improvement is not drastic. Relative to the second-best method, our approach achieves significant enhancements across all metrics (e.g., a 5.8% increase in PSNR, a 2.3% boost in SSIM, a 21.0% leap in LPIPS, a 13.9% advancement in RMSE, and a 0.6% rise in FSIM). The substantial improvements in LPIPS and RMSE underscore our method's capability to refine textures and robustly adapt to various extreme low-light conditions. Fig. 4 showcases a visual comparison on RELLISUR for simultaneous brightness adjustment and ×2 upscaling. The majority of the compared methods suffer from significant noise and blur issues, especially observable in SwinIR, LCUN, and SRFormer. Some cascaded approaches exhibit severe color bias and insufficient brightness enhancement. In contrast, our method is capable of producing images with vivid luminance and excellent restoration of high-frequency structural details. Signal plots intuitively confirm the consistency between our method and the reference images at the pixel intensity level.

**Table 2: Computational efficiency of SOTA methods. Note ✗ indicates the fold increase of the corresponding metric.**

| Method | ❶Parameters (MB) | ❷FLOPs (G) | ❸Inference (S) | ❹FPS |
|---|---|---|---|---|
| Restormer | 26.126 | 35.375 | 0.033 | 26.87 |
| SRFormer | 10.162 | 81.797 | 0.218 | 3.01 |
| HAT | 9.473 | 58.990 | 0.184 | 5.31 |
| Ours* | 1.424₆.₆₅✗ | 20.774₁.₇₀✗ | 0.024₁.₃₇✗ | 41.67₁.₅₅✗ |

**Table 3: Ablation studies of the Alg. 1 (in terms of w/ or w/o S1 and S2) and IHEM on RELLISUR dataset.**

| Config. | Alg. 1 (w/o S1) | Alg. 1 (w/o S2) | w/ IHEM | ❶PSNR↑ | ❷SSIM↑ |
|---|---|---|---|---|---|
| 0 | ✓ | ✓ | ✓ | 22.456 | 0.744 |
| 1 | ✓ | | ✓ | 21.997 ↓0.459 | 0.727 ↓0.017 |
| 2 | | ✓ | ✓ | 22.243 ↓0.213 | 0.735 ↓0.009 |
| 3 | ✓ | ✓ | | 21.959 ↓0.497 | 0.726 ↓0.018 |

**Evaluation on Real Nighttime Scenarios.** We assess the generalization performance of the entire benchmark suite under two real-world challenging scenarios: dimly lit urban streetscapes at night and nocturnal open highway scenes. The quantitative comparison results for both challenging scenarios are presented in Fig. 6, respectively. On the *DarkFace* dataset, we computed three non-paired metrics to evaluate the quantitative scores. Particularly noteworthy is our performance on the MetaIQA metric, where we achieved a 10.5% improvement over the second-best method on DarkFace. This underscores our method's efficacy in effectively restoring a variety of degradations such as noise, blur, and underexposure. Please refer to the *Supplementary Material* for qualitative and quantitative comparisons on the DarkFace dataset. For comparisons on Cityscapes, see the *Supplementary Material* as well.

**Robustness across Diverse Darkness.** Fig. 5 conducts a robustness analysis across varying levels of darkness. Five distinct levels of low exposure are generated by adjusting exposure time, resulting in corresponding dark images (e.g., -2.5EV, -3.0EV, -3.5EV, -4.0EV, and -4.5EV). Our method maintains consistent enhancement across various levels of darkness. This is visually corroborated by the probability density histograms, which demonstrate a uniform consistency distribution across the five different levels of darkness, highlighting the high robustness of our model to inputs under varying levels of darkness. Furthermore, it is noteworthy that the RMSE scores in Tab. 1 also underscore the significant generalization capability of our method across various darkness levels.

**Computational Efficiency.** To evaluate model efficiency, we present the parameters, FLOPs, inference time, and FPS of compared SOTA methods in Tab. 2. The evaluations are performed on a single 2080 Ti GPU using images of size 128 × 128. Excluding the parameters of the frozen SAM model, our network has a parameter count of less than 1.4MB. In conclusion, our network achieves a favorable balance between performance and efficiency.

### 4.4 Ablation Analyses

**Effectiveness of IHEM.** When the IHEM module is removed, as seen in Config.3 of Tab. 3, there is a noticeable performance drop—approximately 2.2% in PSNR and 2.4% in SSIM—compared to the optimal model, Config.0. Fig. 9 presents the ablation results with the feature visualizations facilitated by IHEM. We visualized

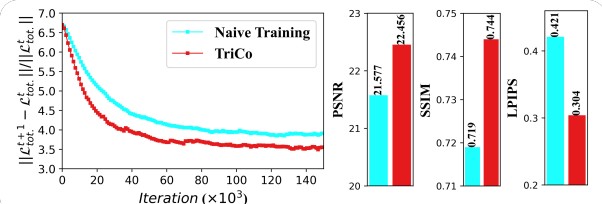

**Figure 8: Comparison analysis of the naive training strategy and our TriCo strategy.**

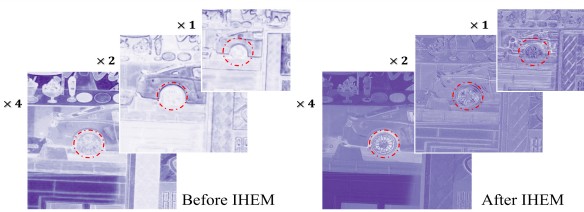

**Figure 9: Illustrating of intermediate layer feature visualization for the IHEM module.**

the features before and after the IHEM process in the last three decoding layers. Upon comparison, the features prior to IHEM appear more sparse and scattered, with a distinct lack of textural detail (see the discernible regions within the dashed circles: the car wheels). Post-IHEM features, however, exhibit a more abstract and semantically rich visual representation. This indicates that IHEM fosters a greater focus on capturing higher-level semantics.

**Analysis of the Solution Algorithm.** We conduct the ablation study to quantify the impact of the proposed algorithm components, with comparative results detailed in Tab. 3, from Config.0 to Config.3. Omitting the S1 strategy alone leads to a performance degradation of approximately 2% in PSNR and 2.2% in SSIM compared to the best-performing model, Config.0. Similarly, removing only the S2 strategy results in a reduction of about 0.9% in PSNR and 1.2% in SSIM relative to Config.0. This delineation underscores the critical importance of synergistically integrating S1 and S2 strategies to achieve the superior performance set forth by Config.0. *Due to space constraints, Supplementary Materials include ablation studies (i.e., LII, LFR, loss functions, etc.).*

## 5 CONCLUSION AND REMARKS

This investigation delves into the intricate realm of brightening and magnifying ultra-dark images, a pursuit fraught with practical complexities due to the dual dilemmas of low resolution and profound darkness. Our tailored TriCo, adopts a tri-level learning strategy that intertwines the tasks of illumination enhancement and super-resolution. By fostering the collaborative learning, TriCo effectively negates the historical deficiencies of isolated or simplistic task handling, yielding superior clarity and artifact reduction.

***Broader Impacts.*** TriCo's strategic innovation extends beyond the ultra-dark super-resolution challenge, advocating for a broader investigation into joint low-level visual and high-level semantic tasks under adverse conditions, which can elevate the development of effective training strategies for a range of coupled tasks.

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
