# OpenReview forum: "Enhancing Images with Coupled Low-Resolution and Ultra-Dark Degradations: A Tri-level Learning Framework"
_acmmm.org/ACMMM/2024/Conference — MM2024 Poster_

### Official Review · Reviewer_Jrum · 2024-05-23

**Rating:** 4
**Confidence:** 3

**Summary:**

The paper introduces TriCo, a Tri-level learning framework for improving visibility and resolution in ultra-dark images. It overcomes current limitations by using a tri-level learning approach, hyper-variables, and some proposed blocks. TriCo outperforms existing methods, making it effective for ultra-dark scenarios.

**Strengths:**

1. The experimental results are good, outperforming existing super-resolution and low-light methods. It demonstrates a robust generalization capability in real-world scenarios.
2. A novel framework is introduced that fosters a collaborative improvement of super-resolution and low-light enhancement tasks.

**Limitations:**

1. The paper writing is hard to follow, and the naming of blocks appears overly complex. For instance, the Integrated Hybrid Expert Modulator is essentially a standard cross attention transformer block, which has been previously utilized in low-light image enhancement research[1].
2. Typically, low-light enhancement models have fewer parameters. The paper employs the large-scale SAM feature for performance improvement but lacks relevant ablation studies. It remains unclear whether the SAM feature usage has significantly contributed to performance enhancement.
3. The tri-level learning framework incorporates an adaptive balance of loss function weights. It would be interesting to explore if other adaptive methods[2] could yield similar results.

[1] Wu, Yuhui, et al. "Learning semantic-aware knowledge guidance for low-light image enhancement." Proceedings of the IEEE/CVF Conference on Computer Vision and Pattern Recognition. 2023.

[2] Sener, Ozan, and Vladlen Koltun. "Multi-task learning as multi-objective optimization." Advances in neural information processing systems 31 (2018).

**Suitability:**

2

---

### Official Review · Reviewer_sCJp · 2024-05-26

**Rating:** 5
**Confidence:** 4

**Summary:**

This paper proposes a new method, namely TriCo, for image joint low-light enhancement and super-resolution. Three technical contributions are claimed.

(i) An Upper (U)-Middle (M)-Lower (L) level nested formulation. The M-L level explicitly delineates the collaborative dependency of two entangled tasks. The U-M level integrates hyper-variables to autonomously enforce positive constraint feedback, thus dismantling the reliance on manual trial-and-error intervention.

(ii) A Phased Gradient-Response (PGR) algorithm as the training mechanism, designed to synergistically optimize
three variables while providing dynamic gradient feedback, thus achieving efficient training and rapid convergence.

(iii) An Integrated Hybrid Expert Modulator (IHEM) that acts as a conduit between illumination prior cues (i.e., intrinsic attributes) and generic semantic model features (i.e., SAM)

**Strengths:**

(i) The writing is clear, good, and easy to follow, especially the mathematical notations in the method part. The presentation is well-addressed. I like the style of the figures in this paper.

(ii) The performance seems solid. The improvement of the proposed TriCo over the state-of-the-art method is about 1 dB. This is very significant. The visual results also demonstrate that the proposed TriCo can reconstruct more high-frequency details and structural contents.

(iii) The ablation study part is also very sufficient to validate the effectiveness of the proposed techniques.

**Limitations:**

(i) The main results are not very convincing. The authors claim that their methods are very effective in high-resolution low-light enhancement. But some important experiments are missing. For example, it would be better if the results on the NTIRE 2024 datasets were provided. The NTIRE 2024 provides 6Kx4K images to evaluate high-resolution low-light enhancement. Also, what about the performance on LOL datasets?

(ii) Many recent highly related works are missing. for example, there are no comparisons with ICCV 2023 works? e.g., Retinexformer [1]?

[1] Retinexformer: One-stage Retinex-based Transformer for Low-light Image Enhancement. In ICCV 2023.

(iii) What are the advantages of joint super-resolution and low-light enhancement? I mean, we can do these two tasks by two methods, one for super-resolution and one for low-light enhancement, right? More explanation about the motivation should be given.

**Suitability:**

2

---

### Official Review · Reviewer_3dMB · 2024-05-27

**Rating:** 5
**Confidence:** 4

**Summary:**

Captured images often suffer from low-resolution and ultra-dark degradation due to device and lighting constraints. Enhancing visibility and resolution simultaneously is crucial, but current methods typically address these tasks separately or through simplistic cascading approaches. They rely heavily on manual loss constraints, leading to compromised efficacy, increased artifacts, and diminished detail fidelity. To overcome these challenges, the authors introduce TriCo, the first Tri-level learning framework. It explicitly considers the bidirectional cooperative relationship between tasks and devises algorithms to address coupled degradation factors. TriCo optimizes across Upper (U)-Middle (M)-Lower (L) levels, modeling synergistic dependencies between illumination learning and super-resolution tasks. Hyper-variables are introduced in the U-M levels to automate constraint learning, reducing reliance on trial-and-error. We employ a Phased Gradient-Response (PGR) algorithm for training, enabling dynamic gradient feedback and rapid convergence. Additionally, the Integrated Hybrid Expert Modulator (IHEM) combines illumination priors with semantic features to guide high-frequency detail recovery at the pixel level.

**Strengths:**

The paper proposes a novel approach to deal with coupled low-resolution and ultra-dark degradations.

The reported results show competitive performance.

**Limitations:**

1.	It is better to add more descriptions in Fig. 3, Fig. 8 and Fig. 9, enabling readers to grasp the content without having to refer to the main text.
2.	The proposed Illumination-Induced Attention is similar with Illumination-Guided Multi-head Self-Attention in Retinexformer.
3.	From the results shown in Figure 6, it appears that the method proposed in this paper does not handle the highlighted areas very well.
4.	The ablation study section lacks in-depth analysis. In addition, it is necessary to include ablation experiments regarding the loss function.

[1] Retinexformer: One-stage retinex-based transformer for low-light image enhancement, CVPR 2023.

**Suitability:**

2

---

### Meta-Review · Area_Chair_iUto · 2024-06-30

**Recommendation:** Accept (Poster)
**Confidence:** 5

**Metareview:**

This submission has been reviewed by three experts, the final ratings from whom are all positive. Considering the quality of the submission, the comments from the reviewers, and the rebuttal from the authors, the paper can be accepted by the conference.